# PROTEGE: Prompt-based Diverse Question Generation from Web Articles

**Vinayak S Puranik**
Amazon
puranikv@amazon.com

**Anirban Majumder**
Amazon
majumda@amazon.com

**Vineet Chaoji**
Amazon
vchaoji@amazon.com

## Abstract

Rich and diverse knowledge-bases (KB) are foundational building blocks for online knowledge sharing communities such as StackOverflow and Quora, and applications such as conversational assistants (aka *chatbots*). A popular format for knowledge bases is question-answer pairs (or FAQs), where questions are designed to accurately match a multitude of queries. In this paper, we address the problem of automatic creation of such Q&A-based knowledge bases from domain-specific, long-form textual content (e.g., web articles). Specifically, we consider the problem of *question generation*, which is the task of generating questions given a paragraph of text as input, with a goal to achieve both ***diversity*** and ***fidelity*** of the generated questions. Towards this goal we propose PROTEGE, a diverse question generation framework which consists of (1) a novel encoder-decoder based Large Language Model (LLM) architecture which can take a variety of prompts and generate a diverse set of candidate questions, and (2) a hill-climbing algorithm that maximizes a sub-modular objective function to balance diversity with fidelity. Through our experiments on three popular public Q&A datasets, we demonstrate that PROTEGE improves diversity by +16% and fidelity by +8% over diverse beam search and prompt-based baselines.

## 1 Introduction

In a data rich era, identifying, extracting and generating responses to user's questions has become the next challenge. While search engines provide a simple interface for users to get responses to their queries, getting answers to complex queries still remains a challenge (Krishna et al., 2021). As a result, specialized knowledge bases that extract and store question-answer pairs have become prevalent.

Many applications rely on a knowledge base of generated question-answer pairs, to ensure reliable, accurate and as close to human-generated information to their users. For instance, a key frustration for online shopping is the difficulty in identifying the right product that suits the requirement. For high-consideration products such as laptops and smartphones, at times customers lack the *human touch* that they would otherwise experience in an offline store where trained sales agents can explain features of each product and provide high-level guidance to select the right one. The sales agent can proactively query the customer to understand her requirement, help refine her needs and finally recommend the right products. In order to bridge the gap between online and offline shopping experience, multi-turn goal-oriented dialog systems, also known as *chatbots* offer a promising direction. Chatbots help users to familiarize technical concepts, acquire domain knowledge, get recommended products that they are likely to buy, closely mimicking the offline shopping experience.

Towards curating a large scale knowledge bank, Large Language Models (LLMs) (Devlin et al., 2019; Raffel et al., 2020; Brown et al., 2020) have shown remarkable success in learning in-depth knowledge from data. They do so without access to any external memory as the knowledge is imbibed in the model parameters. While this is fascinating, on the downside, the model may hallucinate (Marcus, 2020) and generate answers that are factually incorrect. As a result, modern chatbot systems employ a Retrieval-Augmented Generation (RAG) architecture (Lewis et al., 2020) that has two components: a) an encoder-decoder network that does the natural language understanding of user queries and language generation and b) a back-end knowledge-base (KB) that indexes relevant bits of information for the task at hand. The encoder maps user inputs to a dense representation which is used to query the KB and retrieve evidences. The evidences, as well as the input, are fed to the decoder to generate the final response.

Scalable generation of knowledge-bases is fundamental to the success of the RAG and the underly-

ing chatbot application. In this work, we investigate automatic generation of a knowledge-base. Unlike traditional RAG system (Lewis et al., 2020; Izacard et al., 2022) that indexes webpages and large documents, we look for a knowledge-base in the form of question-answer pairs. Not only this helps us improve the accuracy of evidence retrieval, but also allows rich applications to be easily built on top of it e.g., suggesting related questions, navigation of the KB etc, for improved customer experience. Our focus is on educational questions, i.e., questions that help user familiarize with product concepts (*"What is the difference between SSD and HDD?"*) and use-case guidance (*"What is the recommended configuration for a gaming laptop?"*).

Question generation is the task of generating questions given a paragraph of text as input. Question generation quality can be attributed to two characteristics: a) **Fidelity** that measures the semantic coherence of generated questions and our ability to answer them from the input paragraph, and b) **Diversity** which measures lexical and semantic dissimilarity between generated questions. Many previous works (Rajpurkar et al., 2018, 2016; Kwiatkowski et al., 2019) have addressed the task of generating questions from text. While it is essential to generate questions that are of high fidelity, for knowledge-base completion, it is imperative to have a diverse question set. To promote diversity, current question generation models rely on beam search. The resulting set, however contains many structurally similar questions with minor lexical changes that warrant the same answer. There has been prior work (Elhamifar et al., 2012; Song et al., 2018; Vijayakumar et al., 2018) in NLP on diversity. In particular, Song et al. (Song et al., 2018) addresses diversity via Determinantal Point Processes (DPP) for neural conversation models, it can be adapted for question generation task.

While these approaches are helpful in maximizing diversity, they fall short in terms of generating high fidelity output. Naively borrowing these techniques may allow the model to hallucinate and generate questions that are not answerable from the input paragraph. Addressing the task of diverse question generation through the lens of monotone sub-modular function (Bach, 2013) alleviates this problem and provide additional benefits. On one hand, this formulation provides flexibility in controlling diversity and fidelity of the output. On the other hand, we can leverage a well-known greedy

algorithm (Nemhauser et al., 1978) to generate a near optimal set of questions, therefore, increasing yield and quality simultaneously.

We propose **PROTEGE** (**PRO**mp**T**-based div**E**rse question **GE**neration), a diverse question generation framework which consists of two stages (1) a novel encoder-decoder based LLM architecture which can take a variety of prompts and generate a diverse set of candidate questions, and (2) a greedy hill-climbing algorithm that maximizes a sub-modular objective function to balance diversity with fidelity. We demonstrate that PROTEGE improves diversity by +16% and fidelity by +8% while also improving text generation metrics, over strong baselines. Our experiments on three popular public Q&A datasets indicate that PROTEGE consistently outperforms both diverse beam search-based and prompt-based baselines.

## 2 PROTEGE: Prompted Question Generation

Question generation models take a source context $x$ represented as a sequence of sentences $x = (x_1, x_2 \cdots)$, pass them through an encoder to learn its latent representation and finally through a decoder to generate the output question $y = (y_1, y_2 \cdots)$ with one word $y_i$ at a time. Given training data-set $\mathcal{D} = \{(x, y)\}$, the model parameters $\theta$ are learned by maximizing the likelihood function $\sum_{(x,y) \in \mathcal{D}} \log \Pr(y \mid x; \theta)$ The encoder and decoder are implemented as Transformer networks (Vaswani et al., 2017). The encoder consists of $N_e$ layers where each layer contains a self-attention and feed-forward block. The encoder takes an input $x \in \mathbb{R}^{B \times S \times F}$ and passes it through all the encoder blocks to generate an output $h = \text{ENCODER}(x) \in \mathbb{R}^{B \times S \times F}$. [1] The $n^{\text{th}}$ encoder block is a Transformer layer TRANS-FORMER$(x^{(n-1)})$ which takes the input $x^{(n-1)}$ from previous layer and generates the output $x^{(n)}$. The encoder blocks are applied in a sequence and finally we get the output $h = x^{(N_e)}$. To generate the $i^{\text{th}}$ output word $y_{i+1}$, we take the previous words $y_{\cdot j} = y_{1 \cdots i}$ and the encoder output $h$ and pass them through $N_d$ decoder blocks. Each decoder block contains a self-attention, cross-attention and feed-forward layer. The decoder blocks are also applied in a sequence and at the final layer it emits the next word $y_{i+1} = y_{\cdot j}^{(N_d)}$.

---

[1] Here $B$ is the minibatch size, $S$ is the sequence length and $F$ denotes the embedding dimension.

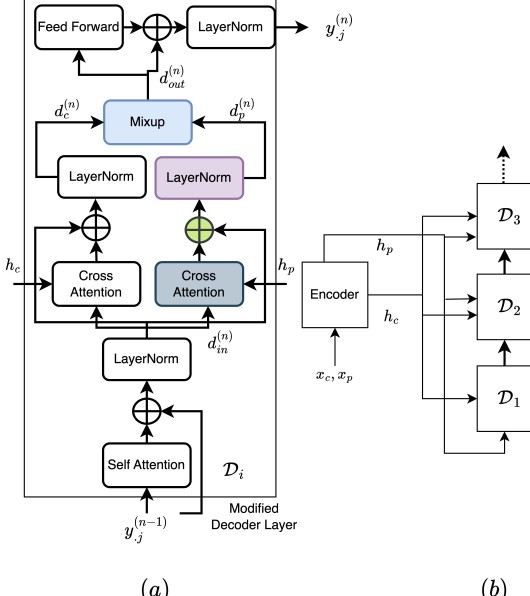

the feed-forward network. More specifically, each decoder layer performs the following computation,

$$d_{in}^{(n)} = \text{LN}\left(y_{.j}^{(n-1)} + \text{SA}(y_{.j}^{(n-1)})\right) \quad (1)$$

$$d_c^{(n)} = \text{LN}\left(d_{in}^{(n)} + \text{CA}(h_c, d_{in}^{(n)})\right) \quad (2)$$

$$d_p^{(n)} = \text{LN}\left(d_{in}^{(n)} + \text{CA}(h_p, d_{in}^{(n)})\right) \quad (3)$$

$$d_{out}^{(n)} = \text{MIXUP}\left(d_c^{(n)}, d_p^{(n)}\right) \quad (4)$$

$$y_{.j}^{(n)} = \text{LN}\left(d_{out}^{(n)} + \text{FF}(d_{out}^{(n)})\right) \quad (5)$$

Here LN, SA, CA, FF are abbreviations for layernorm (Ba et al., 2016), cross-attention (Vaswani et al., 2017), self-attention (Vaswani et al., 2017) and feedforward layers. MIXUP is an aggregation layer that combines the context and prompt cross-attention outputs $d_c^{(n)}, d_p^{(n)}$ to generate $d_{out}^{(n)}$ i.e. $d_{out}^{(n)} = \lambda^T \cdot [d_c^{(n)}, d_p^{(n)}]$. We propose various ways to implement MIXUP: a) treat $\lambda$ as a tunable hyper-parameter, b) learn $\lambda$ as a *free* parameter or via attention (Lin et al., 2017) weights. Note that by setting $\lambda = [1, 0]$, we recover the standard decoder. More details about the architecture choice are described in Appendix A.

## 2.2 Prompt Signals

We use two types of prompts: a) **keyword-based:** we define an entity dictionary based on domain knowledge search keywords such as, brands, features etc. Entities from this dictionary can be used as a prompt, b) **sentence-based:** we identify informative sentences from the context and use them as prompt input. Further, there are two strategies to compute the prompts: a) HEURISTIC: extract prompts from the context based on manually defined rules or ML models, b) ORACLE: extract prompts from both context and the ground-truth question. Note that the ORACLE strategy requires the ground-truth and hence, can be used only during training whereas HEURISTIC can be used during both training and inference. While using ORACLE during training and HEURISTIC in inference leads to train, test mismatch of distribution of prompts, our hypothesis is that it will help establish strong correlation between the prompts and the generated questions. In Appendix B we describe the various prompt signals we have used for our experiments.

Figure 1: Model Architecture of PROTEGE encoder and decoder. The left figure (a) shows the modifications (components shown in color) done to the standard decoder architecture. As shown in the figure (b), the Encoder takes the context and prompt as input and generate representations $h_c$ and $h_p$. The decoder is modified to incorporate cross-attention with $h_c$ and $h_p$ and a mixup layer to aggregate the outputs.

## 2.1 Controlled Generation

For controlled generation of questions, we feed the input document along with various types of prompts to the encoder. This requires some modification to the standard encoder-decoder architecture. We use two encoders: one for the document (or, the context) and the other for the prompt signals (Dou et al., 2021). Similar to Transformer architecture, each encoder has $1 + N_e$ layers where the first $N_e$ layers use shared parameters $\Theta_e$ for the context and prompt. The final encoder layers consist of an additional Transformer block for the context and prompt inputs with individual parameters $\Theta_c, \Theta_p$ respectively. More specifically, given context $x_c$ and prompt $x_p$, we run the following computation on the encoder side,

$$h_c = \text{TRANSFORMER}\left(\text{ENCODER}\left(x_c, \Theta_e\right), \Theta_c\right)$$
$$h_p = \text{TRANSFORMER}\left(\text{ENCODER}\left(x_p, \Theta_e\right), \Theta_p\right)$$

Our decoder attends to both the context and prompt signals $h_c, h_p$. We achieve it by modifying the standard architecture as follows. Unlike standard decoder, each decoder layer attends to both context and prompt embeddings from the encoder via cross-attention layers and their combined output is fed to

## 2.3 Balancing Diversity and Fidelity of Questions

At the end of the first stage of PROTEGE, we have generated a diverse set of questions by varying the prompt input to the model. However, in practice, some of these questions may be irrelevant, i.e., they can't be answered from the current context. In the second stage, we leverage an algorithm that selects a subset of questions which is both relevant and diverse. Let's assume that the previous step has generated $N$ questions $Q = \{q_1, q_2 \cdots, q_N\}$ from a document $D$. The objective of the current step is to select a subset $Q' \subseteq Q$ of size $k$, that maximizes $\zeta(Q', D) =$

$$\eta \cdot \text{diversity}(Q') + (1 - \eta) \cdot \text{relevance}(Q', D) \quad (6)$$

Here $\eta$ is a hyper-parameter that balances the relevance (i.e., fidelity) and diversity. We discuss various choices for implementing the diversity and relevance functions. The relevance of a question set $Q$ is determined via answerability i.e. how likely the question can be answered from the given context. The answerability of a question set $Q'$ is calculated as $f_a(Q', D) = \sum_{q \in Q'} \text{AE}(D, q)$ where AE is an answerability model build on top of standard LLM encoders such as BERT (Devlin et al., 2019). We use n-grams to define diversity of a question bank. Let $z_n(q)$ denote the set of n-grams in $q$ after removing stop-words. We define diversity as diversity$(Q') = \sum_{n \in \{1,2,3\}} | \cup_{q \in Q'} z_n(q) |$.

Note that the diversity expression promotes unique n-grams across questions and has been used as standard metric to measure diversity of text generated by LLMs in prior works, such as (Zhang et al., 2018). It can be noted that the diversity function is sub-modular (Bach, 2013) which makes the objective function $\zeta(Q', D)$ sub-modular as well. Although maximization of a sub-modular function is NP-Hard, it is well-known that the algorithm that greedily picks each item has provably good approximation guarantee (Nemhauser et al., 1978).

## 3 Experiments

### 3.1 Datasets

A supervised dataset for the *question generation* task typically consists of question and answer pairs along with a "context" input. In order to prove the efficacy of our approach for a specific domain of 'shopping guidance', we curate a custom dataset, termed SEARCHQA, by extracting QA pairs from a third-party search engine. We submit customized

shopping guidance queries to the search engine and extract questions, answer snippets and URLs from the search results. We further pre-process the extracted content to form *question, answer, context* triplets. We also leverage three popular benchmark Q&A datasets namely, (1) SQUAD 2.0 (Rajpurkar et al., 2018), (2) NQ Natural Questions dataset (Kwiatkowski et al., 2019), and (3) MS MARCO (Nguyen et al., 2016a). In Appendix C we describe the pre-processing logic used to create the *question, answer, context* triplets from raw datasets. Table 7 in Appendix C lists the dataset statistics.

### 3.2 Implementation details

Our models are based on the popular T5 (Text-to-Text Transfer Transformer) (Raffel et al., 2020) architecture. T5 models closely follow the *encoder-decoder* Transformer implementation originally proposed in (Vaswani et al., 2017) with minor modifications. For baseline models (section 3.3), we use the vanilla `T5ForConditionalGeneration` implementation from the HuggingFace Transformer library (Wolf et al., 2019). For our prompt-based controlled generation models we extend the vanilla implementation by including (as described in section 2.1) (1) an additional encoder for the prompt input which shares parameters with the original encoder, (2) a new cross-attention block in the decoder which is initialized with pre-trained weights from the original cross-attention block.

**Hyper-parameter settings.** To make it feasible to train a large number of models, for all our experiments we use the `t5-small` variant with 60MM parameters as the base implementation. We use a *learning rate* of 5e-5, *epsilon* of 1e-8 with AdamW optimizer. We use a *sequence length* of 512. We train all models up to 10 *epochs* with a training batch size of 4 and choose the checkpoint with the best performance on the validation set. We train our models on a single GPU of an AWS EC2 instance with a GPU memory of 64GB.

### 3.3 Baselines

BASELINE-BEAM Only the context is passed as input (without any additional prompts) and Diverse Beam Search (DBS) (Vijayakumar et al., 2016) is used to generate top-k questions. Diversity parameters *num_beam_groups* and *diversity_penalty* are fine-tuned by optimizing for diversity metrics through a grid search.

BASELINE-PROMPT Prompts are concatenated

| Dataset | Model | Dist-1 ↑ | Dist-2 ↑ | Dist-3 ↑ | Ent-1 ↑ | Ent-2 ↑ | Ent-3 ↑ | BERT Score ↓ | Fidelity ↑ |
|---|---|---|---|---|---|---|---|---|---|
| SEARCHQA | BASELINE-BEAM | 0.5379 | 0.6963 | 0.7675 | 3.5454 | 3.7396 | 3.6759 | 0.9009 | 0.8408 |
| | BASELINE-PROMPT | 0.6333 | 0.7817 | 0.8489 | 3.4667 | 3.6144 | 3.5300 | 0.8895 | 0.8449 |
| | PROTEGE | **0.7351** | **0.9197** | **0.9632** | **3.9869** | **4.2143** | **4.0804** | **0.8235** | **0.9085** |
| SQUAD | BASELINE-BEAM | 0.5589 | 0.6930 | 0.7425 | 3.7763 | 3.9316 | 3.8460 | 0.8763 | 0.8106 |
| | BASELINE-PROMPT | 0.6413 | 0.8215 | 0.8792 | 3.9315 | 4.1840 | 4.1150 | 0.8334 | 0.7958 |
| | PROTEGE | **0.7490** | **0.9340** | **0.9680** | **4.2678** | **4.5285** | **4.4187** | **0.7915** | **0.8812** |
| NQ | BASELINE-BEAM | 0.5032 | 0.5968 | 0.6529 | 3.5582 | 3.6160 | 3.5487 | 0.8991 | 0.7015 |
| | BASELINE-PROMPT | 0.6216 | 0.7522 | 0.8244 | 3.7311 | 3.8580 | 3.8131 | 0.8439 | 0.7447 |
| | PROTEGE | **0.7028** | **0.8522** | **0.9117** | **4.0141** | **4.1824** | **4.1169** | **0.8048** | **0.8151** |
| MS MARCO | BASELINE-BEAM | 0.5148 | 0.6330 | 0.7170 | 2.7588 | 2.6920 | 2.3905 | 0.8986 | 0.7995 |
| | BASELINE-PROMPT | 0.6421 | 0.7756 | 0.8672 | 2.8324 | 2.7905 | 2.5382 | 0.8660 | 0.7502 |
| | PROTEGE | **0.7177** | **0.8693** | **0.9426** | **3.4144** | **3.4548** | **3.2191** | **0.8138** | **0.8278** |

Table 1: Diversity metrics for PROTEGE and baselines across QA datasets.

with context to form a single input to vanilla T5. For a fair evaluation, we use exactly the same set of prompts used in the corresponding controlled generation (PROTEGE) model. For instance, suppose PROTEGE model uses ground-truth question entities as ORACLE prompts (training) and context entities as HEURISTIC prompts (inference), the exact same strategy is used for this baseline as well.

### 3.4 Metrics

**Diversity metrics.** We evaluate on two popular n-gram based *lexical diversity* metrics (1) **Distinct-n** (Li et al., 2016), which measures the percentage of unique n-grams out of total number of n-grams in a set of generated questions. We report Dist-1, Dist-2 and Dist-3 metrics, (2) **Entropy-n** (Zhang et al., 2018), which measures how evenly the n-gram distribution is for a given question set. These two metrics are popularly used in literature to evaluate lexical diversity of generated responses (Zhang et al., 2018; Han et al., 2022; Stasaski and Hearst, 2022; Tevet and Berant, 2020). To measure *semantic diversity* we report a **BERTScore** (Zhang et al., 2020), which is measured as the average BERTScore of each pair of generated questions. BERTScore measures the semantic similarity between a pair of generated sentences, hence lower the average BERTScore better the diversity.

**Fidelity metrics.** To report a fidelity (or "answerability") metric, we train a separate BERT-based model that takes a context and a question and outputs a probability score for the question being answerable from the context. The ROC AUC of this BERT model was observed to be 0.84. We tune the threshold of this model to operate at a precision of 85%, corresponding to a recall of 30%. A higher bar on the precision allows us to select questions which are highly likely to be answerable

from the context, at the cost of missing out on other answerable questions. We compute the answerability score for each generated question and report the average.

**NLG metrics.** Finally, to evaluate the "closeness" of generated questions with respect to the ground-truth questions we also report standard NLG metrics popular in literature, namely: (1) METEOR (Banerjee and Lavie, 2005) which is measured as a harmonic mean of unigram precision and recall, (2) BLEU-4, a cumulative 4-gram BLEU (Papineni et al., 2002) score, which is an evaluation of matching grams of specific order (1-gram, 2-gram etc.) (3) ROUGE-L, a version of ROUGE (Lin and Och, 2004), which measures the longest common subsequence (LCS) between the generated and reference text.

## 4 Results

**Diversity results.** For the SEARCHQA dataset, among several choices for prompt signals (described in section 2.2) we highlight the best results obtained in this section and describe the trade-offs among the choices in section 4.1. In all our tables we **highlight** the first best result and underline the second best result. Note that ↑ for a metric indicates higher values are preferred whereas ↓ indicates lower values are preferred. For SEARCHQA dataset, we observe in table 1 that across all the metrics our algorithm (PROTEGE) does significantly better than both baselines. On Dist-1, Dist-2 and Dist-3 metrics, PROTEGE does 16%, 18%, 13%, respectively, better than the second best result. On Ent-1, Ent-2 and Ent-3 metrics, PROTEGE shows an improvement of 12%, 13% and 11%, respectively, compared to the second best result. PROTEGE also reduces BERTScore by 7%, and improves Fidelity by 8%. PROTEGE generates almost double the

| | | Top-1 | | | Top-2 | | | Top-3 | | |
|---|---|---|---|---|---|---|---|---|---|---|
| Dataset | Model | METEOR | BLEU-4 | ROUGE-L | METEOR | BLEU-4 | ROUGE-L | METEOR | BLEU-4 | ROUGE-L |
| SEARCHQA | BASELINE-BEAM | **0.3088** | **0.3794** | **0.3375** | 0.3598 | **0.4188** | **0.3853** | 0.3968 | 0.4449 | **0.4185** |
| | BASELINE-PROMPT | 0.2850 | 0.3539 | 0.3163 | 0.3359 | 0.3951 | 0.3629 | 0.3567 | 0.4099 | 0.3798 |
| | PROTEGE | 0.2767 | 0.3479 | 0.3004 | **0.3638** | 0.4161 | 0.3803 | **0.4034** | **0.4456** | 0.4161 |
| SQUAD | BASELINE-BEAM | **0.1678** | **0.2346** | **0.2083** | 0.2062 | 0.2674 | 0.2458 | 0.2332 | 0.2900 | 0.2721 |
| | BASELINE-PROMPT | 0.1487 | 0.2109 | 0.1968 | 0.1990 | 0.2583 | 0.2481 | 0.2256 | 0.2830 | 0.2694 |
| | PROTEGE | 0.1593 | 0.2289 | 0.2053 | **0.2155** | **0.2826** | **0.2565** | **0.2490** | **0.3085** | **0.2833** |
| NQ | BASELINE-BEAM | **0.3998** | **0.4634** | **0.4307** | **0.4346** | **0.4884** | **0.4607** | **0.4570** | **0.5035** | **0.4792** |
| | BASELINE-PROMPT | 0.3627 | 0.4333 | 0.4016 | 0.4192 | 0.4725 | 0.4520 | 0.4452 | 0.4888 | 0.4724 |
| | PROTEGE | 0.3089 | 0.3965 | 0.3391 | 0.3855 | 0.4517 | 0.4055 | 0.4262 | 0.4786 | 0.4433 |
| MS MARCO | BASELINE-BEAM | **0.4217** | **0.4712** | **0.4637** | **0.4748** | **0.5105** | **0.5096** | 0.5024 | **0.5298** | **0.5292** |
| | BASELINE-PROMPT | 0.3740 | 0.4228 | 0.4317 | 0.4423 | 0.4799 | 0.4894 | 0.4749 | 0.5048 | 0.5122 |
| | PROTEGE | 0.3788 | 0.4234 | 0.4120 | 0.4631 | 0.4874 | 0.4821 | **0.5064** | 0.5187 | 0.5171 |

Table 2: NLG metrics for PROTEGE and baselines across QA datasets.

number of unique questions compared to BASE-LINE-PROMPT. Thus, given the same context PROTEGE generates a higher number of unique questions which are better both in terms of diversity and fidelity, compared to baselines.

For the benchmark datasets, we observe in Table 1 that across datasets, PROTEGE improves on all the diversity metrics (Dist-n & Ent-n) when compared to both the baselines. For example, on the Dist-1 metric, compared to the second best (which is consistently BASELINE-PROMPT), PROTEGE shows an improvement of 17%, 13% and 12%, respectively, for SQUAD, NQ and MS MARCO. On fidelity, compared to the second best, PROTEGE performs 9%, 9% and 4% better, respectively, for SQUAD, NQ and MS MARCO. Significant reduction is also observed on BERTScore. PROTEGE generates 1 to 3 unique questions (on an average) more than BASELINE-PROMPT.

**NLG results.** We present metrics separately for top-1, top-2 and top-3 generated questions. The metrics for top-k is computed using the question (among top-k) which results in the maximum METEOR score with reference to the ground-truth question. As described earlier, for BASELINE-BEAM we use beam search to generate the top-k questions, while for BASELINE-PROMPT we pick the top-k questions based on generation score. For PROTEGE we select the top-k questions returned by our second-stage algorithm (section 2.3, which greedily selects the question that maximizes diversity and fidelity.

For the SEARCHQA dataset, in table 2 we observe that for the top-1 question the best metrics are obtained from the BASELINE-BEAM model. From our model's point of view this is expected as the topmost question is selected based on diversity and

fidelity objectives, and hence need not be closest to the reference ground-truth. However, as we allow PROTEGE to select more questions (top-2 and top-3) the model often generates a question closer to the ground-truth, which shows in top-2/top-3 results where PROTEGE does better than both the baselines in matching with the reference. In other words, if we allow top-2 questions, PROTEGE shows the best performance with an improvement of 1.1% in METEOR (but, shows second best performance in BLEU-4 and ROUGE-L). Similarly, for top-3 questions the corresponding improvements are +1.7%, +0.2% for METEOR and BLEU-4 scores. We observe similar trends for SQUAD among benchmark datasets.

For the NQ and MS MARCO datasets, although PROTEGE shows a significant improvement over baselines on diversity metrics, improvements are not observed on NLG metrics. We explain our hypothesis for this observation in Appendix D.

**Human evaluation.** We performed human evaluation to compare the quality of top-k generated questions between PROTEGE and BASELINE-BEAM. Annotators were asked to label each set of generated questions (for a given context) w.r.t., a) Readability (*no. of readable & meaningful questions*), b) Diversity (*no. of semantically unique questions*), c) Fidelity (*no. of questions answerable from the context*). In figure 2 we observe that PROTEGE improves on BASELINE-BEAM with an absolute improvement of 5% on readability, 32% on diversity, 36% on answerability. Appendix I describes the details of the human audits.

### 4.1 Ablation studies

**Effect of prompt signals.** For the SEARCHQA dataset we experiment with a variety of (keyword-based and sentence-based) prompt signals as de-

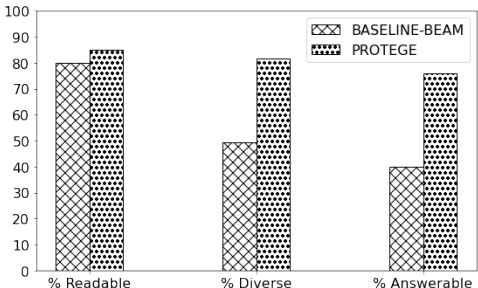

Figure 2: Results of human evaluation.

scribed in 2.2. In table 3 we present the effects of prompt signals on diversity metrics.

| Training prompt (ORACLE) | Inference prompt (HEURISTIC) | Dist-1 ↑ | BERT Score ↓ | Fidelity ↑ |
|---|---|---|---|---|
| Baseline | | 0.5379 | 0.9009 | 0.8408 |
| Answer text | Context span (size=1) | 0.7351 | **0.8235** | **0.9085** |
| Answer text | Context span (size=2) | 0.6925 | 0.8405 | 0.9005 |
| Answer text | Context-span (size=4) | 0.6268 | 0.8730 | 0.8817 |
| Answer keywords | Keywords from context span (size=1) | 0.6931 | 0.8419 | 0.8977 |
| Answer keywords | Keywords from context span (size=2) | 0.5735 | 0.8578 | 0.8906 |
| Answer keywords | Keywords from context span (size=4) | 0.6215 | 0.8829 | 0.8684 |
| Question entities | Context entities | **0.7459** | 0.8234 | 0.8765 |

Table 3: Effect of prompt signals on diversity metrics.

Across all prompt choices, PROTEGE does better than BASELINE-BEAM on all metrics. *Answer text* (with a context span of size 1 during inference) performs the best on BERTScore and Fidelity metrics and second best on Dist-1 metric. *Question entities* shows the best performance on Dist-1 metric, which is due to the fact that the model is trained to generate a question around the specific entity passed as a prompt. Based on these results, we typically use *answer text* as a preferred choice for the prompt. Detailed metrics are in Appendix F.

**Effect of ORACLE prompting.** Across datasets, ORACLE prompting yields the best performance in terms of matching the ground-truth question. (Refer figure 4 and table 11 in Appendix G). This ablation shows the efficacy of our architecture in incorporating the prompt when generating a question, i.e., providing the "exact" prompt elicits a question which is relatively closer to the ground-truth.

**Effect of greedy algorithm.** As described in section 2.3, our algorithm takes the candidate set of questions generated in the first stage (prompt-based controlled generation) and in the second stage performs a greedy algorithm, at each step optimizing

| | Pre-Greedy | | | Post-Greedy | | |
|---|---|---|---|---|---|---|
| Dataset | Dist-1 ↑ | BERT Score ↓ | Fidelity ↑ | Dist-1 ↑ | BERT Score ↓ | Fidelity ↑ |
| SEARCHQA | 0.6158 | 0.8665 | 0.8387 | **0.7351** | **0.8235** | **0.9085** |
| SQUAD | 0.6911 | 0.8161 | 0.8030 | **0.7490** | **0.7915** | **0.8812** |
| NQ | 0.6254 | 0.8414 | 0.7483 | **0.7028** | **0.8048** | **0.8151** |
| MS MARCO | 0.6472 | 0.8402 | 0.7566 | **0.7177** | **0.8138** | **0.8278** |

Table 4: Effect of greedy algorithm on diversity metrics. **Pre-Greedy** is the output of prompt-based controlled generation (section 2.1), while **Post-Greedy** is the output of greedy hill-climbing algorithm (section 2.3).

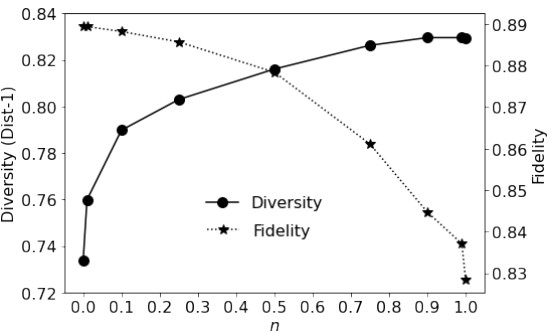

Figure 3: Diversity vs fidelity.

for both diversity and fidelity.

In table 4 we see that as an effect of this greedy algorithm, across datasets both diversity and fidelity metrics show a marked improvement. On an average, post-greedy Dist-1 metric improves by 13% and Fidelity improves by 9%. Further, in Appendix H we show the effects of greedy algorithm on all the diversity and NLG metrics.

**Diversity versus fidelity.** Our algorithm to balance diversity and fidelity of questions (section 2.3) allows us to control the trade-off between diversity and fidelity through the $\eta$ parameter. Figure 3 shows how controlling the $\eta$ parameter allows us to operate at different points for diversity and fidelity. Low $\eta$ results in high fidelity, while high $\eta$ results in high diversity. For our experiments we used an $\eta$ around 0.5 to achieve the right trade-off. Table 8 in Appendix E shows the full metrics as a result of varying $\eta$.

## 4.2 Qualitative study

In Table 5 we provide qualitative examples of questions generated by PROTEGE when compared with BASELINE-BEAM output. Due to paucity of space we do not include the context input which is fed to both the models. The second column shows the output of BASELINE-BEAM given the context input alone. The third column is a sample of the prompts which are fed to PROTEGE model (along with con-

| Context ID | BASELINE-BEAM output | Prompt inputs (keywords and sentences) | PROTEGE output |
|---|---|---|---|
| C1 | Is i3 processor fast? | ram, intel, core i3, processor, ddr3 | How much RAM does Intel Core i3 processor support? |
| | Is i3 processor slow? | The Intel Core i3 processor series supports DDR3 RAM with frequencies of 1,066 or 1,333 MHz. | Can i3 processors support DDR4? |
| | | Core i3 processors have maximum thermal design power ratings ranging from 16 to 73 watts. | What is the maximum power rating of an i3 processor? |
| C2 | Which encoder is best for OBS? | live streaming | What encoder should I use for live streaming? |
| | Is Nvidia good for OBS? | cpu, gpu | Is OBS CPU or GPU intensive? |
| | Which encoder is best for streaming? | gpu, nvidia, graphics card | Is GPU Nvidia graphics card good for OBS? |
| C3 | Is the MacBook Air lighter than M1? | macbook air, weight | Is MacBook Air or M2 heavier? |
| | Is it better to buy MacBook Air or M2? | The M2 Air is a better choice for the vast majority of people over the 13-inch M2 MacBook Pro model, even though the Pro has slightly better performance and longer battery life. | Is the M2 Air better than M2 Pro? |
| | Is the M2 chip better than Pro? | It's also noticed whenever you slot it into a bag or carry it around. | Is the MacBook Air M2 good for traveling? |
| C4 | What is the best bps for streaming? | compression, reducing file sizes, computer science binary, data representation add, bitesize add, useful tool, part | What is the best compression tool for streaming? |
| | What is a good speed for streaming? | user would experience buffering, regular drops, without, quality | Does compressing a file reduce quality? |
| | Is 128kbps good for streaming? | around 1, 500 kbps, sd | How many kbps is normal for streaming? |

Table 5: Table showing anecdotes of questions generated by PROTEGE and BASELINE-PROMPT.

text). We have shown examples of prompt keywords (e.g., first row), as well as prompt sentences (e.g., second row). Finally last column shows the output of PROTEGE model given the prompt and context as input. We observe that given a context PROTEGE leverages the prompts effectively in generating diverse questions when compared to BASELINE-BEAM output. Especially, when sentences are passed as prompts they often appear to be answers to the generated question.

## 5 Related Work

Rule-based (Heilman and Smith, 2010; Fabbri et al., 2020) and DNN based (Sun et al., 2018; Yin et al., 2020) models are used for question generation from text corpora. Answer extraction (Rajpurkar et al., 2016; Kwiatkowski et al., 2019) or machine comprehension (Hermann et al., 2015; Jozefowicz et al., 2016) is a branch of NLP where the goal is to extract answer snippet from text documents given a question as input. In both cases, either the question or the answer is given as input. QA extraction models (Alberti et al., 2019; Du et al., 2017; Reddy et al., 2017; Krishna and Iyyer, 2019) are generally pipeline-based which generates the question and the answer in a sequence. Boros et al. (Boros et al., 2021) uses a question-answering system to detect specific events in textual content (e.g., tweets, blogs). In this context, the entity information is used to frame template-based question (e.g., Where did the [attack] happen?) where attack is an event of interest). Zhang et al. (Zhang et al., 2021) propose combining entity linkage with a QA system. However, our objective is differ-

ent as we enrich the QA extraction technique by augmenting it with entity level metadata.

## 6 Conclusion and Future Work

In this paper we present PROTEGE, a transformer based two-stage question generation framework based on prompts that balances diversity of the generated questions with their fidelity. Through extensive experiments on multiple datasets we show that PROTEGE significantly improves diversity (by +16%) and fidelity (by +8%) compared to strong baselines. As a future work, we will extend our models to simultaneously generate both questions and answers. In preliminary experiments on the task of *extracting answers* for questions from a given context, we have observed that providing the "entities" in the question as additional prompt signals to a BERT-based model improves the answer extraction quality by up to +4.2% in F1 score. Similar applications to other NLG tasks such as document summarization and FAQ creation are possible using the framework proposed in our paper. Extension of our work to non English languages is part of future work.

## Limitations

One limitation of PROTEGE is that it is tightly integrated with existing transformer architecture. Therefore to test its efficacy with Large Language Models (LLMs), we would need access to the pre-trained model parameters. While this is possible for publicly available Large Language Models (LLMs) such as Vicuna (Chiang et al., 2023), Falcon (Penedo et al., 2023) and LLaMA (Tou-

vron et al., 2023), we will miss out state-of-the-art LLMs such as GPT-4 (OpenAI, 2023) and Chat-GPT [2]. Further our approach requires large GPU cluster to train which may lead to higher carbon emission.

Experimental evidences suggest that when context span is used as prompt, our model may hallucinate or mention incomplete product names or product family. For example, instead of "Core i7 12700K CPU", it may generate a question with "Core i7 12700 CPU" which is ambiguous (i7 12700K CPU has a base frequency of 3.6 GHz in comparison to 2.1 GHz for i7 12700F). Generating questions with fully-qualified product names will be a direction of our future work.

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

# APPENDIX

## A  Architecture Choices

The architecture for controlled generation is described in Section 2.1. We once again highlight our key modifications to the standard encoder-decoder architecture in order to achieve our objective of controlled question generation:

- Introduction of an additional encoder stack (with $1 + N_e$ layers) for the "prompt" input. The first $N_e$ layers share the parameters with the encoder stack for the "context" input. The final layer (a transformer block) has its own individual parameters. This choice is not only to reduce computations, but also due to the fact that the difference between context and prompt signals is expected be at a higher-level which is captured only at the top layer of the encoder.

- Modification of each decoder block to attend to the "context" and "prompt" embedding via separate cross-attention layers. The output of the cross-attention blocks are combined using a MIXUP operation and fed to the feed-forward layer. It is important to note that this is a fundamental change to the standard decoder architecture. The intuition for this architecture is that the cross-attention with the prompt signal influences the decoder to focus on certain topics while the cross-attention with the context influences the decoder to improve relevance of the questions generated w.r.t. the context. Together the aggregated cross-attention influences the model to generate a question around the prompt while remaining true to the information present in the context. This aggregation of the decoder cross-attention with the context and prompt embed-

ding happens throughout all the decoder layers.

One of our baselines, BASELINE-PROMPT, can be viewed as a combination of "context" and "prompt" tokens at the input level (i.e., *early fusion*) via cross-attention. We also performed limited experiments to combine the encoder outputs for the context ($h_c$) and prompt ($h_p$) (i.e., *mid-level fusion*) and observed it to perform worse than BASELINE-PROMPT. Our experiments also suggested that a single cross-attention and MIXUP was not sufficient to guarantee faithfulness of the generated questions w.r.t. the prompt signal, and hence needs to be repeated in each decoder layer.

## B  Prompt signals

In table 6 we describe the various prompt signals we have used for our experiments.

## C  Dataset details

**Search QA (SEARCHQA):** We start by creating a set of custom templates for shopping guidance queries (e.g., *things to consider when buying a <category>*, *main features of a <usecase> <category>*). We expand the templates by populating the slots to create a seed set of search queries (e.g., *main features of a gaming laptop*). For each query, we submit a search request to a third-party search engine and extract questions, answer snippets and URLs. Further, we extract the textual content from the URLs. We thus collect a dataset of size ~100K. After filtering out rows where we are (a) unable to extract the URL content (b) unable to locate the answer in the extracted URL content we are left with ~60K datapoints. Finally, for each *question, answer, URL* datapoint, starting from the textual content extracted from the URL we extract multiple paragraphs (each containing the answer in different locations) to create a "context" input (data augmentation). Thus, each *question, answer, URL* datapoint expands into N *question, answer, context* datapoints. From the URLs set we create two mutually exclusive set of domain names, one each for *train* and *test* datasets (to ensure that models generalize across unseen domains), which allows us to create a training dataset with ~100K rows, and a dev and test datasets each with ~5K rows each.

**SQuAD 2.0(SQUAD):** The Stanford Question Answering Dataset 2.0 (Rajpurkar et al., 2016) is a public dataset consisting of crowdsourced questions on a selection of Wikipedia articles. The dataset consists of a paragraph/context, a set of questions relevant to the context and for each question, an answer which is a phrase from within the context. We ignore unanswerable questions (where *is_impossible = True*). We split the original train dataset into train (~80K rows) and dev (~5K rows) by splitting based on *titles*. We sample from the original validation set (consisting of ~10K unique datapoints) to create a test set (~5K rows).

**Google Natural Questions (NQ):** Natural Questions (Kwiatkowski et al., 2019) is a collection of real user questions submitted to Google and answers gathered from Wikipedia by annotators. From the original dataset we parse the *question_text*, the *long_answer* (and treat it as a context) and the *short_answer* (and treat it as an answer) which is usually a phrase from within the context (except for yes/no answers). We filter out contexts that contain HTML tables (<Table>) and also filter out very long contexts (>= 20 sentences). We sample from the original training data of ~307K datapoints to create a train (~120K rows) and dev (~5K rows). We similarly parse the original ~7.8K validation set to create a test set (~3.5K rows).

**MS MARCO:** MS MARCO (Microsoft Machine Reading Comprehension) (Nguyen et al., 2016b) is a large scale collection of datasets (machine reading comprehension, passage ranking, etc.) of which we leverage the question answering dataset. Queries (questions) are sampled from Bing logs and 10 most relevant passages for the query are generated. Human annotators then tag passages that contain an answer to the question and identify the answers from the relevant passages. From the original dataset, for a given query we randomly select one answer and then randomly sample 3 passages (selecting one passage that contains the answer and two passages that do not contain the answer), shuffle and concatenate the passages to form our input context. We sample from the original train and dev datasets to create a train (~100K rows), dev (~5K rows) and test (~5K rows).

## D  NQ & MS MARCO observations

For the NQ and MS MARCO datasets, although PROTEGE shows a significant improvement over baselines on diversity metrics, improvements are not observed on NLG metrics. In the case of NQ and MS MARCO datasets, the answer is often a short phrase (specifically, in NQ we use the "short answer" provided in the dataset). During inference,

|  | Training (ORACLE) | Inference (HEURISTIC) |
|---|---|---|
| **Answer text** | Text of the ground-truth answer. | Iteratively select a window of k (=2) sentences from the context. |
| **Answer keywords** | Keywords derived from the ground-truth answer using RAKE (Rapid Automatic Keyword Extraction) algorithm. | Iteratively select a window of k (=2) sentences from the context and derive the keywords from the context window. |
| **Question entities** | Entities from ground-truth question identified using a pre-defined dictionary of domain-specific entities. | Iteratively select a window of k (=2) sentences from the context and derive the entities from the context window. |

Table 6: Prompt signals.

|  | SEARCHQA | | | SQUAD | | | NQ | | | MS MARCO | | |
|---|---|---|---|---|---|---|---|---|---|---|---|---|
|  | **Train** | **Dev** | **Test** | **Train** | **Dev** | **Test** | **Train** | **Dev** | **Test** | **Train** | **Dev** | **Test** |
| **# of rows** | 100000 | 5432 | 5432 | 82775 | 4022 | 5000 | 124011 | 5000 | 3438 | 100000 | 5000 | 5000 |
| **Avg. context size (# sentences)** | 10.00 | 10.00 | 10.00 | 5.12 | 4.85 | 5.38 | 4.18 | 4.23 | 4.16 | 10.18 | 9.69 | 9.67 |
| **Avg. question size (# words)** | 7.17 | 7.21 | 7.19 | 10.04 | 10.73 | 10.27 | 9.05 | 9.06 | 9.09 | 5.97 | 5.88 | 5.88 |
| **Avg. answer size (# sentences)** | 2.12 | 2.15 | 2.15 | 1.00 | 1.00 | 1.00 | 0.79 | 0.80 | 0.77 | 1.15 | 1.15 | 1.16 |
| **Avg. answer size (# words)** | 39.22 | 39.70 | 39.87 | 3.17 | 3.06 | 3.61 | 3.39 | 3.41 | 3.42 | 12.91 | 14.81 | 15.32 |

Table 7: Dataset statistics.

we select the top-k phrases/keywords from the context, using an unsupervised keyword detection algorithm (RAKE), as prompt signals. Hence, for most examples the specific phrase/keyword which is part of the ground-truth answer does not always get selected. Due to this reason, although we generate a number of answerable and diverse questions from the context, we may not necessarily generate a question semantically similar to the ground-truth question. On the other hand, the baseline techniques take as input only the context and no other controlling signal and hence, are more likely to generate a question similar to the ground-truth. This issue is not seen with SEARCHQA and SQUAD datasets, where answers are usually complete sentences which are picked as a candidate prompt signals during inference. Further we note that for all datasets, including NQ and MS MARCO, when the model is provided with the exact ground-truth answer-based prompt, it indeed generates a question semantically closer to the ground-truth question as observed in the Section 4.1 ("Effect of OR-ACLE prompting").

Below we provide examples to illustrate why on NQ and MS MARCO datasets the performance of PROTEGE is lower than baselines on NLG metrics. As seen in the examples below the question generated by PROTEGE depends on the phrase/keyword that gets selected during inference, which need not

necessarily match the phrase/keyword that elicits ths ground-truth question.

Example in NQ:
**Context**: *The American Civil War was fought in the United States from 1861 to 1865...*
**Ground-truth question**: *who took part in the american civil war*
**Ground-truth short answer**: *nationalists of the Union*

| Prompt | Generated question |
|---|---|
| *president abraham lincoln* | *who was president when the civil war began* |
| *united states* | *where did the civil war take place in* |

Example in MS MARCO:
**Context**: *Jesse James (VII) Producer | Actor. At first glance, Jesse James is the consummate biker rebel...*
**Ground-truth question**: *who is married to jesse james*
**Ground-truth short answer**: *Karla James, Janine Lindemulder, Sandra Bullock and Alexi DeJoria.:*

# E   Ablation: Balancing diversity vs fidelity

Table 8 shows the full metrics as a result of varying $\eta$.

| Prompt | Generated question |
|---|---|
| *Jesse James (VII) Producer \| Actor.* | *who is jesse james* |
| *Tattoos, knives, goatee, black t-shirts and skulls all around him* | *what is jesse james famous for* |

| eta | Dist-1 | Dist-2 | Dist-3 | Ent-1 | Ent-2 | Ent-3 | Fidelity |
|---|---|---|---|---|---|---|---|
| **0.00** | 0.5686 | 0.7338 | 0.8149 | 3.5514 | 3.7571 | 3.7190 | **0.8895** |
| **0.01** | 0.5901 | 0.7599 | 0.8383 | 3.6003 | 3.8137 | 3.7683 | 0.8894 |
| **0.10** | 0.6165 | 0.7898 | 0.8633 | 3.6668 | 3.8854 | 3.8288 | 0.8882 |
| **0.25** | 0.6287 | 0.8030 | 0.8743 | 3.6984 | 3.9183 | 3.8569 | 0.8857 |
| **0.50** | 0.6408 | 0.8161 | 0.8850 | 3.7303 | 3.9518 | 3.8854 | 0.8784 |
| **0.75** | 0.6508 | 0.8263 | 0.8928 | 3.7580 | 3.9794 | 3.9084 | 0.8611 |
| **0.90** | 0.6542 | **0.8296** | **0.8952** | 3.7683 | 3.9890 | 3.9164 | 0.8447 |
| **0.99** | 0.6547 | **0.8296** | 0.8951 | **3.7709** | **3.9911** | **3.9183** | 0.8371 |
| **1.00** | **0.6554** | 0.8295 | 0.8944 | 3.7651 | 3.9825 | 3.9077 | 0.8284 |

Table 8: Diversity vs Fidelity with varying $\eta$.

## F Ablation: Effect of prompt signals

In tables 9 and 10 we present the complete set of diversity and NLG metrics based on the choice of prompt signals. Specifically for the NLG metrics, regardless of the choice of prompt signal for top-1 question, BASELINE-BEAM generates questions closest to ground-truth followed by *answer keywords*. For top-2 and top-3, the best strategy in general is to pass *answer keywords* as prompts during training and keywords from context spans of size 2 or 4 during inference. The best result with *answer keywords* is better than the best result with *answer text*, indicating that model benefits more when passed keywords rather than full text as guidance signal. Among different context span sizes passing a larger window (2 or 4 sentences) leads to better results. The worst performing is *question entities* possibly because model tends to overfit on the specific prompted entities, while it generalizes when passed with a larger window of keywords/sentences.

## G Ablation: ORACLE prompting

Figure 4 and table 11 (with the complete set metrics for ORACLE vs HEURISTIC prompting) shows that on an average there is a 40+% improvement in METEOR metrics from HEURISTIC prompting compared to ORACLE prompting.

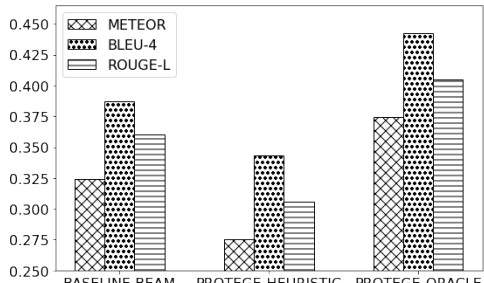

Figure 4: ORACLE vs HEURISTIC prompts.

## H Ablation: Effect of greedy algorithm

Table 12 shows the effect of greedy algorithm on the full set of diversity metrics. As expected, across all datasets diversity metrics improve with greedy algorithm.

In table 13 we observe that post-greedy top-1 METEOR reduces for some datasets. This is expected as the generated question from the first stage is often replaced by a question which displays high diversity and fidelity. However, at top-2 and top-3 the METEOR slightly increases (except for NQ) indicating that the greedy algorithm implicitly favors the question more closer to the ground-truth (which is also expected to be answerable) as long as it improves the diversity.

## I Audit SOP

We perform human audits with 2 auditors to compare the generated questions between PROTEGE and BASELINE-BEAM (for a sample of ~200 contexts). For each data point, auditors record the following details regarding the top-k generated questions: (A) Are the questions readable and meaningful (i.e., well-formed and complete sentences)? (B) Out of the readable questions, how many questions are semantically unique (measures semantic diversity)? (C) Out of the readable questions, how many questions are answerable from the context (measures fidelity)? In case of conflicts on any of the labels, a third auditor re-verifies the decision to resolve the conflict. Finally, we take a cumulative count for each aspect and measure the percentage of readable, unique and answerable questions.

| Dataset | Model | Training prompt (ORACLE) | Inference prompt (HEURISTIC) | Dist-1 ↑ | Dist-2 ↑ | Dist-3 ↑ | Ent-1 ↑ | Ent-2 ↑ | Ent-3 ↑ | BERT Score ↓ | Fidelity ↑ |
|---|---|---|---|---|---|---|---|---|---|---|---|
| SEARCHQA | BASELINE-BEAM | NA | NA | 0.5379 | 0.6963 | 0.7675 | 3.5454 | 3.7396 | 3.6759 | 0.9009 | 0.8408 |
| | PROTEGE | Answer-Text | Context span (size=1) | 0.7351 | 0.9197 | **0.9632** | 3.9869 | 4.2143 | **4.0804** | **0.8235** | **0.9085** |
| | | Answer-Text | Context span (size=2) | 0.6925 | 0.8829 | 0.9390 | 3.8926 | 4.1388 | 4.0343 | 0.8405 | 0.9005 |
| | | Answer-Text | Context-span (size=4) | 0.6268 | 0.8077 | 0.8786 | 3.6813 | 3.9115 | 3.8432 | 0.8730 | 0.8817 |
| | | Answer-Keywords | Keywords from context span (size=1) | 0.6931 | 0.8810 | 0.9376 | 3.9216 | 4.1666 | 4.0688 | 0.8419 | 0.8977 |
| | | Answer-Keywords | Keywords from context span (size=2) | 0.6552 | 0.8410 | 0.9059 | 3.8174 | 4.0624 | 3.9861 | 0.8578 | 0.8906 |
| | | Answer-Keywords | Keywords from context span (size=4) | 0.6215 | 0.7859 | 0.8572 | 3.6352 | 3.8325 | 3.7730 | 0.8829 | 0.8684 |
| | | Question-Entities | Context entities | **0.7459** | **0.9278** | 0.9629 | **3.9967** | **4.2152** | 4.0664 | 0.8234 | 0.8765 |

Table 9: Effect of prompts on diversity metrics.

| Dataset | Model | Training prompt (ORACLE) | Inference prompt (HEURISTIC) | Top-1 | | | Top-2 | | | Top-3 | | |
|---|---|---|---|---|---|---|---|---|---|---|---|---|
| | | | | METEOR | BLEU-4 | ROUGE-L | METEOR | BLEU-4 | ROUGE-L | METEOR | BLEU-4 | ROUGE-L |
| SEARCHQA | BASELINE-BEAM | NA | NA | **0.3088** | **0.3794** | **0.3375** | 0.3598 | 0.4188 | 0.3853 | 0.3968 | 0.4449 | **0.4185** |
| | PROTEGE | Answer-Text | Context span (size=1) | 0.2444 | 0.3186 | 0.2740 | 0.3273 | 0.3845 | 0.3510 | 0.3779 | 0.4223 | 0.3966 |
| | | Answer-Text | Context span (size=2) | 0.2588 | 0.3334 | 0.2902 | 0.3430 | 0.3992 | 0.3671 | 0.3921 | 0.4341 | 0.4110 |
| | | Answer-Text | Context-span (size=4) | 0.2821 | 0.3560 | 0.3102 | 0.3655 | 0.4200 | **0.3866** | 0.3964 | 0.4440 | 0.4148 |
| | | Answer-Keywords | Keywords from context span (size=1) | 0.2646 | 0.3387 | 0.2895 | 0.3520 | 0.4060 | 0.3697 | 0.4002 | 0.4415 | 0.4157 |
| | | Answer-Keywords | Keywords from context span (size=2) | 0.2767 | 0.3479 | 0.3004 | 0.3638 | 0.4161 | 0.3803 | **0.4034** | **0.4456** | 0.4161 |
| | | Answer-Keywords | Keywords from context span (size=4) | 0.2938 | 0.3627 | 0.3162 | **0.3666** | **0.4201** | 0.3836 | 0.3929 | 0.4398 | 0.4072 |
| | | Question-Entities | Context entities | 0.2275 | 0.2929 | 0.2502 | 0.3004 | 0.3534 | 0.3191 | 0.3387 | 0.3786 | 0.3521 |

Table 10: Effect of prompts on NLG metrics.

| Dataset | Prompt Choice (Training) | BASELINE-BEAM | | | PROTEGE (HEURISTIC) | | | PROTEGE (ORACLE) | | |
|---|---|---|---|---|---|---|---|---|---|---|
| | | METEOR top-1 | BLEU-4 top-1 | ROUGE-L top-1 | METEOR top-1 | BLEU-4 top-1 | ROUGE-L top-1 | METEOR top-1 | BLEU-4 top-1 | ROUGE-L top-1 |
| SEARCHQA | Answer-Text | 0.3088 | 0.3794 | 0.3375 | 0.2821 | 0.3560 | 0.3102 | 0.3221 | 0.4043 | 0.3518 |
| | Answer-Keywords | | | | 0.2938 | 0.3627 | 0.3162 | 0.3107 | 0.3915 | 0.3353 |
| | Question-Entities | | | | 0.2275 | 0.2929 | 0.2502 | 0.4213 | 0.5032 | 0.4564 |
| SQUAD | Answer-Text | 0.1678 | 0.2346 | 0.2083 | 0.1593 | 0.2289 | 0.2053 | 0.2777 | 0.3442 | 0.3133 |
| NQ | Answer-Text | 0.3998 | 0.4634 | 0.4307 | 0.3089 | 0.3965 | 0.3391 | 0.4224 | 0.4763 | 0.4518 |
| MS MARCO | Answer-Text | 0.4217 | 0.4712 | 0.4637 | 0.3788 | 0.4234 | 0.4120 | 0.4940 | 0.5364 | 0.5226 |

Table 11: Effect of ORACLE prompting.

| Dataset | Pre-Greedy | | | | | Post-Greedy | | | | |
|---|---|---|---|---|---|---|---|---|---|---|
| | Dist-1 ↑ | Dist-2 ↑ | Dist-3 ↑ | BERT Score ↓ | Fidelity ↑ | Dist-1 ↑ | Dist-2 ↑ | Dist-3 ↑ | BERT Score ↓ | Fidelity ↑ |
| SEARCHQA | 0.6158 | 0.8009 | 0.8743 | 0.8665 | 0.8387 | **0.7351** | **0.9197** | **0.9632** | **0.8235** | **0.9085** |
| SQUAD | 0.6911 | 0.8744 | 0.9222 | 0.8161 | 0.8030 | **0.7490** | **0.9340** | **0.9680** | **0.7915** | **0.8812** |
| NQ | 0.6254 | 0.7597 | 0.8294 | 0.8414 | 0.7483 | **0.7028** | **0.8522** | **0.9117** | **0.8048** | **0.8151** |
| MS MARCO | 0.6472 | 0.7984 | 0.8928 | 0.8402 | 0.7566 | **0.7177** | **0.8693** | **0.9426** | **0.8138** | **0.8278** |

Table 12: Effect of greedy algorithm on diversity metrics.

| Dataset | Pre-Greedy | | | Post-Greedy | | |
|---|---|---|---|---|---|---|
| | METEOR top-1 | METEOR top-2 | METEOR top-3 | METEOR top-1 | METEOR top-2 | METEOR top-3 |
| SEARCHQA | **0.2952** | **0.3646** | 0.4002 | 0.2767 | 0.3638 | **0.4034** |
| SQUAD | 0.1386 | 0.1996 | 0.2368 | **0.1593** | **0.2155** | **0.2490** |
| NQ | **0.3550** | **0.4146** | **0.4435** | 0.3089 | 0.3855 | 0.4262 |
| MS MARCO | 0.3662 | 0.4562 | 0.5021 | **0.3788** | **0.4631** | **0.5064** |

Table 13: Effect of greedy algorithm on NLG metrics.