# OpenReview forum: "PROTEGE: Prompt-based Diverse Question Generation from Web Articles"
_EMNLP/2023/Conference — EMNLP 2023 Findings_

### Official Review · Reviewer_aVs7 · 2023-08-04

**Soundness:** 4

**Excitement:**

3: Ambivalent: It has merits (e.g., it reports state-of-the-art results, the idea is nice), but there are key weaknesses (e.g., it describes incremental work), and it can significantly benefit from another round of revision. However, I won't object to accepting it if my co-reviewers champion it.

**Paper Topic And Main Contributions:**

This paper proposes a method named Prompt-based diverse question generation (PROTEGE), which enables diverge question generation through a decoder modification and greedy hill-climbing algorithm. Specifically, the method involves forwarding the prompt to the different cross-attention layers in the decoder layer and then combining the representations from two cross-attention layers. The experimental results show that the proposed method can generate questions with improved fidelity and diversity compared to previous prompting methods that simply concatenate the prompt to the input context.

**Questions For The Authors:**

- Is the paper formatted according to the EMNLP guidelines? There appear to be inconsistencies in the paper format, such as the font.
- In lines 215-216, are the context and prompt encoded separately, or is there an alternative approach?
- Have the authors tested their method on other controllable generation tasks, or is the proposed approach solely limited to question generation tasks? It would be interesting to explore the method's applicability in other scenarios.

**Reasons To Accept:**

- The experimental results are promising, and the paper conducts a thorough analysis of the results in terms of diversity and fidelity. The inclusion of human evaluations in Figure 2 supplements the reliability of the findings.
- The idea of employing separate cross-attention architecture for conditional generation is a novel contribution to my knowledge.

**Reasons To Reject:**

- The proposed method shows inferior performance against baselines when evaluated using NLG metrics (METEOR, BLEU, ROUGLE). It would be beneficial to include additional examples in the Appendix, explaning why tehse scores are relatively low.
- Some important details are missing in the experiment setting. For instance, the paper does not specify the number of examples used in human evaluation or the number of participants involved in the evaluation process. Furthermore, there are concerns about the reliability of the fidelity metrics based on the trained neural network. The suitability of the trained BERT model for measuring the fidelity score of generated questions is questionable, considering results in Table 2.

**Reproducibility:**

3: Could reproduce the results with some difficulty. The settings of parameters are underspecified or subjectively determined; the training/evaluation data are not widely available.

**Reviewer Confidence:**

2: Willing to defend my evaluation, but it is fairly likely that I missed some details, didn't understand some central points, or can't be sure about the novelty of the work.

**Typos Grammar Style And Presentation Improvements:**

- Table 5, C1-Baseline output is missing

---

> ### Author Rebuttal · Authors · 2023-08-29
>
> Dear Reviewer, we thank you for your valuable comments. Please find below our response for the feedback and questions you have shared:
>
> &nbsp;
> ***
> *“The proposed method shows inferior performance against baselines when evaluated using NLG metrics (METEOR, BLEU, ROUGLE). It would be beneficial to include additional examples in the Appendix, explaning why tehse scores are relatively low.”*
>
> We agree with the reviewer that PROTEGE shows inferior performance against baselines on NLG metrics (shown in Table 2), especially for the NQ and MS MARCO datasets. For SEARCHQA and SQUAD datasets, PROTEGE shows a marked improvement over baseline techniques on NLG metrics, for the case of Top-2 and Top-3 generated questions (please check METEOR scores for Top-2 and Top-3 in Table 2). However for the Top-1 case, PROTEGE performance is lower than baselines. Please note that **Table 1 (Diversity metrics)** and **Table 2 (NLG metrics)** are two different dimensions of results, as explained below.
>
> **Diversity vs NLG metrics:** Diversity metrics demonstrates the ability of PROTEGE to generate a diverse set of answerable (relevant) questions from a given context. NLG metrics (METEOR, BLEU-4, ROUGE-L) demonstrate how close the generated questions are to the ground-truth question. Our goal in this paper is not to generate the ground-truth question alone accurately, but to generate a set of highly diverse questions which are answerable from the given context. Hence, NLG metrics are not our primary choice of metrics. However, we still report NLG metrics to demonstrate that among our top-k generated questions, at least one of the questions is semantically similar to the ground-truth question. Table 2 shows that the top-1 question generated by PROTEGE, while being highly relevant (answerable), need not necessarily be similar to the ground-truth question for the given context (hence the lower performance). When considering top-2 or top-3 questions, PROTEGE starts generating at least one question closer to the ground-truth which is shown in the NLG performance being better than BASELINE-BEAM (for SEARCHQA and SQUAD datasets). However, in the case of NQ and MS-MARCO even at top-2/top-3 the NLG metrics are not better than baseline, which can be explained with the following hypothesis.
>
> **NLG metrics on NQ, MS MARCO:** In the case of NQ and MS MARCO datasets, the answer is often a short phrase (specifically, in NQ we use the “short answer” given in the dataset). During inference, as the prompt signal we select the top-k phrases/keywords using an unsupervised keyword detection algorithm (RAKE). Hence, it is quite likely that for most examples we do not select the specific phrase/keyword which is part of the ground-truth answer. Due to this reason, although we generate a number of answerable and diverse questions for the context, we may not necessarily generate a question semantically similar to the ground-truth question. On the other hand, the baseline technique takes as input only the context and no other controlling signal and hence, is more likely to generate a question similar to the ground-truth.
>
> **Additional examples for NQ, MS MARCO:** The examples below for NQ and MS MARCO further illustrate our explanation. Please note, although the generated questions are relevant and answerable, they are different from the ground-truth question. As suggested by the reviewer, we will add these examples to the Appendix in the final version of the paper. We thank the reviewer for this suggestion.
>
> | **Dataset**               | **NQ**                                                                        |
> |---------------------------|-------------------------------------------------------------------------------|
> | **Context**               | _The American Civil War was fought in the United States from 1861 to 1865..._ |
> | **Ground-truth question** | _who took part in the american civil war_                                     |
> | **Ground-truth answer**   | _nationalists of the Union_                                                   |
> | **Prompt-1**              | _president abraham lincoln_                                                   |
> | **Generated question-1**  | _who was president when the civil war began_                                  |
> | **Prompt-2**              | _united states_                                                               |
> | **Generated question-2**  | _where did the civil war take place in_                                       |
>
> | **Dataset**               | **MS MARCO**                                                                                              |
> |---------------------------|-----------------------------------------------------------------------------------------------------------|
> | **Context**               | _Jesse James (VII) Producer \| Actor. At first glance, Jesse James is the consummate biker rebel..._      |
> | **Ground-truth Question** | _who is married to jesse james_                                                                           |
> | **Ground-truth Answer**   | _Karla James, Janine Lindemulder, Sandra Bullock and Alexi DeJoria._                                      |
> | **Prompt-1**              | _Jesse James (VII) Producer \| Actor._                                                                    |
> | **Generated Question-1**  | _who is jesse james_                                                                                      |
> | **Prompt-2**              | _Tattoos, knives, goatee, black t-shirts and skulls all around him_ |
> | **Generated Question-2**  | _what is jesse james famous for_                                                                          |
>
> **NLG metrics with Oracle prompting:** Finally we also note that, for all datasets, including NQ and MS MARCO, when the model is provided the exact ground-truth answer-based prompt, it indeed generates a question semantically closer to the ground-truth question as described in the Section “Effect of ORACLE prompting” and the NLG metrics shown in Figure 4.
>
> &nbsp;
> ***
> *“Some important details are missing in the experiment setting. For instance, the paper does not specify the number of examples used in human evaluation or the number of participants involved in the evaluation process.”*
>
> We thank the reviewer for pointing out this important detail which may require further information. We had 2 auditors to work on a sample of ~200 data points for BASELINE-BEAM and PROTEGE outputs. For each data point (context and a set of generated questions), auditors were asked to label 3 values a) Readability (no. of questions that are meaningful & well-formed) b) Diversity (no. of semantically unique questions which elicit different answers from the context) c) Fidelity (no. of questions that are answerable from the given context). In case of conflicts on any of the labels, a 3rd auditor re-verified the decision to resolve the conflict. The Standard Operating Procedure (SOP) followed for labeling by auditors is described in Appendix F (Audit SOP).
>
> &nbsp;
> ***
> *"Furthermore, there are concerns about the reliability of the fidelity metrics based on the trained neural network. The suitability of the trained BERT model for measuring the fidelity score of generated questions is questionable, considering results in Table 2."*
>
> In the earlier point, we have provided the key differences between diversity metrics (from Table 1) and NLG metrics (from Table 2) and explained our hypothesis of why PROTEGE shows inferior performance on NLG metrics for some of the datasets. The reliability of the fidelity metrics mentioned by the reviewer is an important point that needs further information.
>
> In order to report the answerability metrics, we train a standard BERT-based model that takes a context and a question as input, and predicts the probability of the question being answerable from the context. We leverage this score to select the question (from the candidate set of questions output from the first stage of PROTEGE) that not only adds maximum diversity to the already selected pool, but also is most answerable from the given passage (based on its answerability score). The ROC AUC of this BERT model is 0.84. We tune the threshold of this model to operate at a precision of 85%, corresponding to a recall of ~30%. A higher bar on the precision allows us to select questions which are highly likely to be answerable from the context, at the cost of losing other answerable questions. **The reliability of this answerability model is also demonstrated through human audits (line 463) where PROTEGE was found to improve upon the baseline by 36% on the fidelity score**.
>
> &nbsp;
> ***
> **Questions For The Authors-1:**
> *“Is the paper formatted according to the EMNLP guidelines? There appear to be inconsistencies in the paper format, such as the font.”*
>
> Yes, the reviewer is correct to point out that the paper is formatted using Computer Modern Roman font, which is one of the approved formats for EMNLP and has been approved by the Program Chairs. However, in the final version, we will recompile the paper using the Times font which is the more popular format.
>
> &nbsp;
> ***
> **Questions For The Authors-2:**
> *“In lines 215-216, are the context and prompt encoded separately, or is there an alternative approach?”*
>
> As indicated by the reviewer, the “context” and the “prompt” signals are encoded separately. We have introduced an additional encoder stack (with $1 + N_e$ layers) for the “prompt” input, where the first $N_e$ layers share their parameters with the encoder for the “context” input and a final layer (a transformer block) with individual parameters. This is to reduce the computations and also due to the fact that the difference between context and prompt signals will be at a higher-level which is captured only at the top layer of the encoder. Also this allows our model to learn unique representations for the “context” (a longer data with multiple concepts/topics) and the “prompt” (shorter, more specific signal such as entities, specific answer etc.). The decoder attends to the “context” and “prompt” embeddings separately via cross-attention layers and aggregates the representations through a MIXUP operation. These details are described in Section 2.1 of our paper. Alternatively, one of our baselines, BASELINE-PROMPT, can be thought of combining “context” and “prompt” tokens at the input level (early fusion) via cross-attention. We also did limited experiments (not presented in the paper) to combine the encoder output of the “context” embeddings and “prompt” embeddings (mid-level fusion) and observed it to perform worse than BASELINE-PROMPT.
>
> &nbsp;
> ***
> **Questions For The Authors-3:**
> *“Have the authors tested their method on other controllable generation tasks, or is the proposed approach solely limited to question generation tasks? It would be interesting to explore the method's applicability in other scenarios.”*
>
> We agree that our experiments in this paper are focused mainly on the Question Generation task. Our intention was to show the efficacy of the proposed technique strongly on a given NLG task by including experiments on a diverse variety of QA datasets (SEARCHQA and SQUAD, NQ, MS MARCO). However, our proposals including the architectural framework and the greedy hill-climbing algorithm are generic enough to be applied on a wide variety of NLG tasks. Although not included in this paper, we have applied the technique to the task of answer extraction from a given document (context) and question. In our experiments, we have observed that providing the “entities” in the question as additional prompt signals to a BERT-based model improves the answer extraction quality by up to **+4.2% in F1 score**. Similar applications to other NLG tasks such as document summarization and FAQ creation are possible using the framework proposed in our paper.
>
> &nbsp;
> ***
> *"Table 5, C1-Baseline output is missing"*
>
> In Table 5, C1 example the baseline model had output only two unique questions. Hence, the table shows only two outputs for baseline in C1.

---

### Official Review · Reviewer_ypQJ · 2023-08-10

**Soundness:** 4

**Excitement:**

4: Strong: This paper deepens the understanding of some phenomenon or lowers the barriers to an existing research direction.

**Paper Topic And Main Contributions:**

The authors propose PROTEGE - a question generation method based on the LLM architecture and based on input prompts. In addition, this study also introduces the greedy hill-climbing algorithm to balance the fidelity and diversity of the generated question.

**Reasons To Accept:**

This study introduced a new method (PROTEGE) to generate questions based on LLM architecture. The study performed various experiments on three datasets and provided a lot of insight into the results. The article can open a period of generating Q&A data more automatically and faster than previous methods.

**Reasons To Reject:**

The study has not shown practical applications to help people for this research.
We want more empirical diversity (add more languages if possible)
And what is the way to make sure the questions that are generated stick to the passage and are answerable?

**Reproducibility:**

4: Could mostly reproduce the results, but there may be some variation because of sample variance or minor variations in their interpretation of the protocol or method.

**Reviewer Confidence:**

3: Pretty sure, but there's a chance I missed something. Although I have a good feel for this area in general, I did not carefully check the paper's details, e.g., the math, experimental design, or novelty.

---

> ### Author Rebuttal · Authors · 2023-08-29
>
> Dear Reviewer, we thank you for your valuable comments. Please find below our response for the feedback and questions you have shared:
>
> &nbsp;
> ***
> *“The study has not shown practical applications to help people for this research. We want more empirical diversity (add more languages if possible)”*
>
> We agree that our experiments in this paper are focused exclusively on the Question Generation task. Our intention was to show the efficacy of the proposed technique strongly on a given NLG task by including experiments on a diverse variety of QA datasets (SEARCHQA and SQUAD, NQ, MS MARCO). However, our proposals including the *architectural framework* and the *greedy hill-climbing algorithm* are generic enough to be applied on a wide variety of NLG tasks. Although not included in this paper, we have applied the technique to the task of answer extraction from a given document (context) and question. In our experiments, we have observed that providing the “entities” in the question as additional prompt signals to a BERT-based model improves the answer extraction quality by up to **+4.2% in F1 score**. Similar applications to other NLG tasks such as *document summarization* and *FAQ creation* are possible using the framework proposed in our paper. Extension of our work to non English languages is part of future work.
>
> &nbsp;
> ***
> *“what is the way to make sure the questions that are generated stick to the passage and are answerable?”*
>
> We have addressed the point about generating questions that are relevant to the context through our “fidelity” objective function and the corresponding metric. Fidelity is part of the objective function that we optimize for in the greedy hill-climbing algorithm (referred to as *relevance* in equation 6), and is also a metric that we report results for. Fidelity means answerability, i.e., whether the generated question is answerable from the given passage/context. Given a context and a question, we have trained a BERT model that outputs an answerability score. We leverage this score to select the question (from the candidate set of questions output from the first stage of PROTEGE) that not only adds maximum diversity to the already selected pool, but also is most answerable from the given passage (based on its answerability score). This process naturally avoids questions which do not stick to the passage or are irrelevant to the given context. As a result, as shown in Table 1, PROTEGE outperforms the baseline techniques on the Fidelity metric (last column) across datasets. More fundamentally, since our training data comprises of ground-truth questions that are answerable from a context, our modelling framework tends to generate questions that are relevant to the given context.

---

### Official Review · Reviewer_MjJZ · 2023-08-11

**Typos Grammar Style And Presentation Improvements:** 1. In Figure 1, using identifiers (a)…
**Soundness:** 3

**Excitement:**

2: Mediocre: This paper makes marginal contributions (vs non-contemporaneous work), so I would rather not see it in the conference.

**Paper Topic And Main Contributions:**

This paper mainly studies how to improve the prompt-based question generation from web articles, called PROTEGE,  especially under the environment of domain-specific and long-form textual content. To achieve this, the paper proposes the encoder-decoder based LLM which can take diverse prompt, consisting of keyword-based and sentence-based, and generates question. In addition, to get the balance between diversity and fidelity, a hill-climbing algorithm with sub-modular objective function is leveraged. In this method, the diversity is calculated by the number of unique n-grams and the relevance is calculated with the answerability with standard LLM. The experiment results are evaluated with diverse datasets and analyzed with diversity metrics and fidelity metrics.

**Reasons To Accept:**

1. Easy-to-read and well-written paper.
2. Analyzed through various experiments and metrics.
3. Study about the relevance between diversity and fidelity.


**Reasons To Reject:**

1. A relatively simple framework structure. The key difference from existing question generation models is the addition of an extra layer at the end of the encoder stage to extract embedding of prompt. Furthermore, these extracted embeddings are utilized in the decoder stage through cross-attention. However, applying attention across different input modalities is one of the commonly used approaches in NLP.

2. Lack of detail explanation. In Section 2.2, a detail explanation of heuristic method is missing. In Table 4, it is unclear the meaning of pre-greedy and post-greedy.

3. Insufficient explanation about limited performance. In table 2, the NLG metrics for fidelity are worse in NQ and MS MARCO dataset. It needs more analysis.

**Reproducibility:**

2: Would be hard pressed to reproduce the results. The contribution depends on data that are simply not available outside the author's institution or consortium; not enough details are provided.

**Reviewer Confidence:**

4: Quite sure. I tried to check the important points carefully. It's unlikely, though conceivable, that I missed something that should affect my ratings.

---

> ### Author Rebuttal · Authors · 2023-08-29
>
> Dear Reviewer, we thank you for your valuable comments. Please find below our response for the feedback and questions you have shared:
>
> &nbsp;
> ***
> *“A relatively simple framework structure. The key difference from existing question generation models is the addition of an extra layer at the end of the encoder stage to extract embedding of prompt. Furthermore, these extracted embeddings are utilized in the decoder stage through cross-attention. However, applying attention across different input modalities is one of the commonly used approaches in NLP."*
>
> We thank the reviewer for this insightful comment. We would like to highlight the following key modifications made to the standard encoder-decoder architecture in order to achieve our objective of controlled question generation:
>
> * Introduction of an additional encoder stack (with $1 + N_e$ layers) for the “prompt” input. The first $N_e$ layers share the parameters with the encoder stack for the “context” input. The final layer (a transformer block) has its own individual parameters. This is to reduce the computations and also due to the fact that the difference between context and prompt signals will be at a higher-level which is captured only at the top layer of the encoder.
> * We modify each decoder block to attend to the “context” and “prompt” embeddings via separate cross-attention layers. The output of the cross-attention blocks are combined using a MIXUP operation (Figure 1) and fed to the feed-forward layer. It is important to note that this is a fundamental change to the standard decoder architecture. The intuition behind this architecture is that the cross-attention with the “prompt” signal influences the decoder to focus on certain topics while the cross-attention with the “context” influences the decoder to improve relevance of the questions generated w.r.t the context. Together the aggregated cross-attention influences the model to generate a question around the “prompt” while remaining true to the information present in the context. This aggregation of the decoder cross-attention with the “context” and “prompt” embeddings happens throughout all the decoder layers.
>
> The reviewer is correct in pointing out that applying cross-attention across different modalities (“context” and “prompt”) is similar to a single application of the MIXUP operation.  One of our baselines, BASELINE-PROMPT, can be thought of combining “context” and “prompt” tokens at the input level (early fusion) via cross-attention. We also did limited experiments (not presented in the paper) to combine the encoder output of the “context” embeddings and “prompt” embeddings (mid-level fusion) and observed it to perform worse than BASELINE-PROMPT.  Our experiments also suggested that a single cross-attention and MIXUP is not sufficient to guarantee faithfulness of the generated questions w.r.t the prompt and hence, we propose the need for cross-attention and MIXUP to be repeated in each decoder layer.
>
> &nbsp;
> ***
> *“Lack of detail explanation. In Section 2.2, a detail explanation of heuristic method is missing”*
>
> In Section 2.2 we have mentioned HEURISTIC as a class of rule-based methods for generating prompt signals, and we have mentioned the exact choice of the heuristics separately in Section 4.1 (“Effect of prompt signals”) as an implementation detail. Below we provide the definitions of training prompt signals (along with the corresponding inference prompts) used in Table 3 of our paper:
>
> **A) Answer text:**
>
> * For training: Text of the ground-truth answer.
> * For inference: Iteratively select a window of k (=2) sentences from the context.
>
> **B) Answer keywords:**
>
> * For training: Keywords derived from the ground-truth answer using RAKE (Rapid Automatic Keyword Extraction) algorithm.
> * For inference: Iteratively select a window of k (=2) sentences from the context and derive the keywords from the context window
>
> **C) Question entities:**
>
> * For training: Entities from ground-truth question identified using a pre-defined dictionary of domain-specific entities.
> * For inference: Iteratively select a window of k (=2) sentences from the context and derive the entities from the context window.
>
> &nbsp;
> ***
> *“In Table 4, it is unclear the meaning of pre-greedy and post-greedy.”*
>
> We apologize for any confusion regarding the terminology of ‘pre-greedy’ and ‘post-greedy’. We have defined the greedy algorithm in Section 2.3 via Equation 6. Further, we have explained the setup of the experiments in Section “Effect of greedy algorithm” (Lines 501 to 513) and presented the results in Table 4. Please consider the additional explanation below:
>
> Our approach (PROTEGE) works in two stages. **In the first stage,** we generate a candidate set of questions using prompt-based controlled generation (described in Section 2.1) using the architecture defined in Figure 1. For a given context, we vary the prompt signals (using heuristic rules) to generate a collection of a variety of questions. **In the second stage,** we implement a *greedy hill-climbing algorithm* that optimizes for equation 6 (i.e., greedily choose a set of generated questions from the first stage that achieves the best balance between n-gram diversity and fidelity). In Table 4, we call the output of the first stage as **‘pre-greedy’** and the output of the second stage as **‘post-greedy’**. We show the effect of this greedy hill-climbing algorithm by demonstrating that the resulting set of questions post the application of greedy algorithm has an improved *diversity* (by 13%) and *fidelity* (by 9%), as a direct effect of optimizing for the two criteria.
>
> &nbsp;
> ***
> *“Insufficient explanation about limited performance. In table 2, the NLG metrics for fidelity are worse in NQ and MS MARCO dataset. It needs more analysis.”*
>
> The reviewer has rightly called out that in Table 2, the NLG metrics are inferior to baselines for NQ and MS MARCO datasets. Table 2 shows the NLG metrics (METEOR, BLEU-4, ROUGE-L) across 4 datasets. Fidelity metrics are reported as part of Table 1 (last column) wherein PROTEGE shows significant improvement, including for NQ and MS MARCO datasets. Regarding the performance of NQ and MS MARCO on NLG metrics, we’d like to share the following hypothesis:
>
> In the case of NQ and MS MARCO datasets, the answer is often a short phrase (specifically, in NQ we use the “short answer” given in the dataset). During inference, as the prompt signal we select the top-k phrases/keywords using an unsupervised keyword detection algorithm (RAKE). Hence, it is quite likely that for most examples we do not select the specific phrase/keyword which is part of the ground-truth answer. Due to this reason, although we generate a number of answerable and diverse questions for the context, we may not necessarily generate a question semantically similar to the ground-truth question. On the other hand, the baseline technique takes as input only the context and no other controlling signal and hence, is more likely to generate a question similar to the ground-truth. This issue is not seen with SEARCH-QA and SQUAD datasets where answers are usually complete sentences and during inference we pick each sentence in the context as a candidate prompt signal. The **examples below from NQ and MS MARCO** further illustrate our explanation.
>
> | **Dataset**               | **NQ**                                                                        |
> |---------------------------|-------------------------------------------------------------------------------|
> | **Context**               | _The American Civil War was fought in the United States from 1861 to 1865..._ |
> | **Ground-truth question** | _who took part in the american civil war_                                     |
> | **Ground-truth answer**   | _nationalists of the Union_                                                   |
> | **Prompt-1**              | _president abraham lincoln_                                                   |
> | **Generated question-1**  | _who was president when the civil war began_                                  |
> | **Prompt-2**              | _united states_                                                               |
> | **Generated question-2**  | _where did the civil war take place in_                                       |
>
> | **Dataset**               | **MS MARCO**                                                                                              |
> |---------------------------|-----------------------------------------------------------------------------------------------------------|
> | **Context**               | _Jesse James (VII) Producer \| Actor. At first glance, Jesse James is the consummate biker rebel..._      |
> | **Ground-truth Question** | _who is married to jesse james_                                                                           |
> | **Ground-truth Answer**   | _Karla James, Janine Lindemulder, Sandra Bullock and Alexi DeJoria._                                      |
> | **Prompt-1**              | _Jesse James (VII) Producer \| Actor._                                                                    |
> | **Generated Question-1**  | _who is jesse james_                                                                                      |
> | **Prompt-2**              | _Tattoos, knives, goatee, black t-shirts and skulls all around him_ |
> | **Generated Question-2**  | _what is jesse james famous for_                                                                          |
>
> Further we note that, for all datasets, including NQ and MS MARCO, when the model is provided the exact ground-truth answer-based prompt, it indeed generates a question semantically closer to the ground-truth question as described in the Section **“Effect of ORACLE prompting”** and the NLG metrics shown in Figure 4.
>
> &nbsp;
> ***
> *“In Figure 1, using identifiers (a) and (b) for sub-figures can make a more precise explanation.”*
>
> We thank the reviewer for this suggestion. In the final version of the paper we will add identifiers (a) and (b) for the sub-figures of Figure 1, for a better explanation.
>
> &nbsp;
> ***
> *“This paper makes marginal contributions (vs non-contemporaneous work)”*
>
> In this paper, we have proposed a novel modification to the fundamental architecture of a decoder layer to achieve the objective of conditional generation. In particular, the prompt signal is encoded separately (from the context signal) which the decoder attends to and combines the representations with the vanilla cross-attention throughout all the layers. This helps the decoder gradually achieve the desired combination of generating an answerable question from the context, while grounding it around concepts provided as prompt signals. In order to leverage the most from this architecture, we have proposed a greedy hill-climbing algorithm that optimizes for a combination of diversity (so that multiple unique questions are generated) and fidelity (so that questions stay true to the context provided). We have applied our technique to a unique setting of question generation which is not among the most widely discussed topics in literature (unlike popular NLG tasks such as document summarization). Finally, although not presented in this paper, we have applied our technique to the task of answer extraction (given context and a question) and demonstrated that providing entities as prompt signals improves the extraction quality by up to **+4.2% in F1 score** over a baseline that doesn't incorporate the prompt signal.
>
> &nbsp;
> ***
> *“The contribution depends on data that are simply not available outside the author's institution or consortium; not enough details are provided.”*
>
> We have demonstrated the efficacy of PROTEGE through experiments on SQUAD, NQ and MS MARCO which are popular benchmark Q&A datasets that are publicly available. The post-processing details for these datasets are described in detail in Appendix A (“Dataset details”). We will also make the post-processed versions of these public datasets available as part of the final version of the paper. Similarly, the SEARCHQA dataset is created from publicly available sources and we will release this dataset as part of the final version of the paper. Our implementation details, including hyper-parameter settings are described in Section 3.2. We believe these steps will help in reproducing the claims made in our paper.

---

### Meta-Review · Area_Chair_K6BD · 2023-09-15

**Recommendation:** 4

**Metareview:**

This paper introduces PROTEGE, a novel technique for question generation over web articles. The basic concept is to condition generation on a prompt, either keyword-based (e.g. an entity in the article) or phrase-based (e.g. a sentence in the article). Cross-attention to this prompt is then used during generation, and different prompts can be selected to induce diversity. The authors combine this with a hill-climbing submodule that allows control over the tradeoff between diversity and fidelity.

The experiments in the paper test the model using common QA datasets. The authors demonstrate that both fidelity and – especially – question diversity benefits from their approach. That is, given an article, the questions generated by PROTEGE are significantly better at covering the aspects of the article compared to prior work. The reliability of these findings are supplemented with human experiments, providing additional backing.

The paper is well-written and clearly presented, and the empirical findings are a convincing demonstration of PROTEGE's efficacy.

---

### Meta-Review · Area_Chair_pKQY · 2023-09-15

**Recommendation:** 4

**Metareview:**

This paper presents a method called PROTEGE that does a two-step process for question generation from text sources.  First, a set of diverse questions is generated, using a pair of encoders one of the document and the other for the "prompt signals".  Next, there is a submodular objective function defined by a pair of quantities meant to control the trade-off between diversity and fidelity, and this optimized and hill-climbed when arriving at the final set of questions.

The authors could have done a slightly better job motivating this work.  One of the reviewers noted the lack of practical application.  However, this work not only pertains to the claimed area of constructing QA datasets, but also to the related—and very practical—areas of building better QA systems, and/or evaluating them.  This came out, albeit in a slightly oblique way, in the rebuttal to reviewer ypQJ.  Nevertheless, I believe this paper warrants inclusion in the conference.

---

### Decision · Program_Chairs · 2023-10-07

**Decision:**

Accept-Findings

**Comment:**

This paper introduces PROTEGE, a novel technique for question generation over web articles. The basic concept is to condition generation on a prompt, either keyword-based (e.g. an entity in the article) or phrase-based (e.g. a sentence in the article). Cross-attention to this prompt is then used during generation, and different prompts can be selected to induce diversity. The authors combine this with a hill-climbing submodule that allows control over the tradeoff between diversity and fidelity.

The experiments in the paper test the model using common QA datasets. The authors demonstrate that both fidelity and – especially – question diversity benefits from their approach. That is, given an article, the questions generated by PROTEGE are significantly better at covering the aspects of the article compared to prior work. The reliability of these findings are supplemented with human experiments, providing additional backing.

The paper is well-written and clearly presented, and the empirical findings are a convincing demonstration of PROTEGE's efficacy.|This paper presents a method called PROTEGE that does a two-step process for question generation from text sources.  First, a set of diverse questions is generated, using a pair of encoders one of the document and the other for the "prompt signals".  Next, there is a submodular objective function defined by a pair of quantities meant to control the trade-off between diversity and fidelity, and this optimized and hill-climbed when arriving at the final set of questions.

The authors could have done a slightly better job motivating this work.  One of the reviewers noted the lack of practical application.  However, this work not only pertains to the claimed area of constructing QA datasets, but also to the related—and very practical—areas of building better QA systems, and/or evaluating them.  This came out, albeit in a slightly oblique way, in the rebuttal to reviewer ypQJ.  Nevertheless, I believe this paper warrants inclusion in the conference.